# Imaging immunomodulatory treatment responses in a multiple sclerosis mouse model using hyperpolarized $^{13}$C metabolic MRI

Caroline Guglielmetti [1,2✉], Christian Cordano [3], Chloé Najac [4], Ari J. Green[3,5] & Myriam M. Chaumeil [1,2✉]

## Abstract

**Background** In recent years, the ability of conventional magnetic resonance imaging (MRI), including $T_1$ contrast-enhanced (CE) MRI, to monitor high-efficacy therapies and predict long-term disability in multiple sclerosis (MS) has been challenged. Therefore, non-invasive methods to improve MS lesions detection and monitor therapy response are needed.

**Methods** We studied the combined cuprizone and experimental autoimmune encephalomyelitis (CPZ-EAE) mouse model of MS, which presents inflammatory-mediated demyelinated lesions in the central nervous system as commonly seen in MS patients. Using hyperpolarized $^{13}$C MR spectroscopy (MRS) metabolic imaging, we measured cerebral metabolic fluxes in control, CPZ-EAE and CPZ-EAE mice treated with two clinically-relevant therapies, namely fingolimod and dimethyl fumarate. We also acquired conventional $T_1$ CE MRI to detect active lesions, and performed ex vivo measurements of enzyme activities and immunofluorescence analyses of brain tissue. Last, we evaluated associations between imaging and ex vivo parameters.

**Results** We show that hyperpolarized [1-$^{13}$C]pyruvate conversion to lactate is increased in the brain of untreated CPZ-EAE mice when compared to the control, reflecting immune cell activation. We further demonstrate that this metabolic conversion is significantly decreased in response to the two treatments. This reduction can be explained by increased pyruvate dehydrogenase activity and a decrease in immune cells. Importantly, we show that hyperpolarized $^{13}$C MRS detects dimethyl fumarate therapy, whereas conventional $T_1$ CE MRI cannot.

**Conclusions** In conclusion, hyperpolarized MRS metabolic imaging of [1-$^{13}$C]pyruvate detects immunological responses to disease-modifying therapies in MS. This technique is complementary to conventional MRI and provides unique information on neuroinflammation and its modulation.

## Plain language summary

Magnetic resonance imaging (MRI) is widely used in the clinic to diagnose multiple sclerosis (MS), which affects the central nervous system and leads to a range of disabling symptoms. However, MRI is often not capable of detecting how well a patient responds to therapies, in particular those targeting the immune system. We questioned whether an advanced MRI method called hyperpolarized $^{13}$C MRS could help. Using a mouse model for MS, we showed that hyperpolarized $^{13}$C MRS can detect response to two therapies used in the clinic, namely fingolimod and dimethyl fumarate when conventional MRI could not. We also showed that this method is sensitive to the immune response. As hyperpolarized $^{13}$C MRS is becoming available in many centers worldwide, it could be used to evaluate existing and new treatments for people living with MS, improving care and quality of life.

[1] Department of Physical Therapy and Rehabilitation Science, University of California San Francisco, San Francisco, CA, USA. [2] Department of Radiology and Biomedical Imaging, University of California San Francisco, San Francisco, CA, USA. [3] Department of Neurology, Weill Institute for Neurosciences, University of California at San Francisco, San Francisco, CA, USA. [4] Department of Radiology, C.J. Gorter MRI Center, Leiden University Medical Center, Leiden, The Netherlands. [5] Department of Ophthalmology, University of California at San Francisco, CA San Francisco, USA. ✉email: caroline.guglielmetti@ucsf.edu; myriam.chaumeil@ucsf.edu

Multiple sclerosis is an immune-mediated demyelinating disease of the central nervous system (CNS) with prominent adaptive immune cell infiltration of the brain and spinal cord and activation of innate immune processes as well[1]. Existing therapies to treat the disease principally target the adaptive immune system and either mitigate the migration, proliferation, or activation state of lymphocytes that participate in immune-mediated injury or deplete populations of immune cells believed important in the pathogenesis of the disease. Effects on the innate system from most of these therapies are secondary or indirect—but may be important in the clinical benefits seen with treatment. In the clinic, assessment of therapeutic response relies on measuring the frequency of relapses, evaluating for accumulation of disability, and monitoring brain and spinal cord lesions with anatomical magnetic resonance imaging (MRI)[2]. Importantly, the predictive value of standard clinical MRI to evaluate long-term multiple sclerosis disability and therapy efficacy has been recently challenged[3]. Furthermore, these clinical and standard imaging measures are only indirectly linked to the immunological phenomena that drive disease. Although $T_1$ contrast-enhanced MRI following injection of gadolinium-based contrast agents enables the detection of disruptions in blood–brain-barrier (BBB) integrity at the level of the tight junction[4] and thereby differentiate between active and inactive lesions, there is currently no clinically available MRI method able to measure immune cell activity in normal-appearing tissue, thus hampering direct monitoring of therapeutic response.

[18F]-fluorodeoxyglucose ([18F]FDG) positron emission tomography (PET) imaging can detect inflammation[5] in other tissues, but high background uptake in the brain limits its use for studying CNS inflammation. Radiotracers binding to the translocator protein 18 kDa (TSPO) have shown some potential for reflecting microglial activation but lack cellular specificity. This is unsurprising as TSPO may be upregulated not only by immune cells but also by glial and endothelial cells[6]. Further, a TSPO polymorphism has been associated with various tracer binding affinities, thus requiring genetic testing to determine patient eligibility[7]. Over the past few years, novel tracers have shown promise as a means of detecting neuroinflammatory processes in neurological disorders and MS and may represent useful probes in the near future[6,8]. However, the high cost and dependence on the availability of a cyclotron may be a limiting factor for repeated clinical evaluation of the same individual.

Hyperpolarized $^{13}$C magnetic resonance spectroscopy (MRS) is an emerging method that enables direct in situ assessment of metabolic activity in living organisms, and its utility to detect cancerous lesions and normal brain metabolism has been demonstrated in humans[9,10]. Recently, metabolic imaging of immune cells has been made possible using hyperpolarized $^{13}$C MRS in vitro and in vivo in various organs including the heart, liver, joints, and brain[11–20]. Specifically, hyperpolarized [1-$^{13}$C] pyruvate enables the detection of increased [1-$^{13}$C]lactate production from highly glycolytic pro-inflammatory activated mononuclear phagocytes and T lymphocytes[12,16–18,20,21]. Furthermore, we have previously shown that increased [1-$^{13}$C]pyruvate to [1-$^{13}$C]lactate flux is associated with the presence of pro-inflammatory innate immune cells in a toxin-induced model of MS[12] and in a moderate traumatic brain injury model[11]. Yet, it remains to be elucidated whether hyperpolarized $^{13}$C MRS could also provide a measure of immunotherapeutic effectiveness in a non-oncological context, in any organs, including the brain, thus addressing the unmet clinical need for direct in situ evaluation of immune cell activity and response to therapy.

Although the experimental autoimmune encephalomyelitis (EAE) model is the most widely used model to study the inflammatory aspects of MS[22,23], the pathology mostly affects the spinal cord, optic nerve, and cerebellum, not the whole brain. Therefore, standard EAE is less suitable to validate imaging methods aiming at visualizing MS pathology in brain lesions. The cuprizone (CPZ) model is a well-established model[24], in which CPZ, a copper chelator, is added to the animal diet to induce oligodendrocytes death and demyelination. The location and timing of CPZ-induced lesions are highly reproducible, making this model extremely useful to validate imaging tools. However, a limitation of this model is that demyelination is not mediated by T cells[25], and few T cells are present in CPZ-induced lesions. The combined CPZ and EAE (CPZ-EAE) model is a relatively recent model of inflammatory demyelination[26,27], characterized by reproducible induction of wide-spread demyelinated brain lesions that present parenchymal infiltration of T cells, increased number of microglia and macrophages, and reactive astrocytes, which are key features of MS pathology. Besides recapitulating key cellular aspects of MS in the brain in a reproducible manner, the CPZ-EAE model displays spinal cord pathology resulting in physical impairment linked to disease progression. Altogether, these features make it an attractive model to investigate the potential of hyperpolarized [1-$^{13}$C]pyruvate to detect pathological changes and evaluate the effect of immunomodulatory treatments.

In the healthy brain, hyperpolarized [1-$^{13}$C]pyruvate crosses the BBB and enters cells through monocarboxylate transporters, where it is further metabolized[28]. Brain imaging studies using hyperpolarized [1-$^{13}$C]pyruvate report subsequent detection of downstream metabolites including hyperpolarized [1-$^{13}$C]lactate and [$^{13}$C]bicarbonate[10,29,30]. In MS, however, the BBB may be disrupted, potentially influencing the delivery of hyperpolarized [1-$^{13}$C]pyruvate[31]. Hyperpolarized [$^{13}$C]urea has been described as an agent with high potential to evaluate perfusion in models of prostate cancer, heart, kidney, brain, and spinal cord[13,32–40]. As such, co-injection of hyperpolarized [1-$^{13}$C]pyruvate with hyperpolarized [$^{13}$C]urea represents a unique way to evaluate changes in metabolism as well as probe delivery in the MS brain. As hyperpolarized [$^{13}$C]urea is currently under investigation in clinical studies, we evaluated its potential to detect neurological alterations and response to therapies in the CPZ-EAE model.

Here, we show that the flux of hyperpolarized [1-$^{13}$C]pyruvate towards lactate is decreased in vivo in the CPZ-EAE MS model of brain inflammatory demyelination following dimethyl fumarate (DMF) and fingolimod (FTY720) treatments. These two immunomodulatory therapies differ by their primary mechanism of action as FTY720 prevents CCR7+ lymphocytes' egress from lymphoid tissues[41], and DMF attenuates inflammatory responses, possibly activates pathways that contribute to oxidative stress/burst, and restricts the proliferation of certain lymphocyte subsets[42,43]. Modulation of the hyperpolarized [1-$^{13}$C]pyruvate flux towards lactate is associated with changes in innate immune cells levels, pyruvate dehydrogenase activity (PDH), pyruvate dehydrogenase kinase 1 (PDK1) expression, and disease symptoms. Hyperpolarized [$^{13}$C]urea levels remain unchanged regardless of inflammatory demyelination and therapeutic responses. Gadolinium-enhanced MRI is associated with ex vivo markers of BBB breakdown and disease burden across animals. Interestingly, gadolinium-enhanced MRI detects a response to FTY720 treatment but does not detect the effect of DMF therapy. Altogether our findings demonstrate that hyperpolarized [1-$^{13}$C] pyruvate MRS imaging provides a way to detect inflammatory lesions and responses to therapies, which would otherwise be missed when relying solely on gadolinium-enhanced MRI.

## Methods

**Animal model induction and immunotherapy treatment.** All animal research was approved by the Institutional Animal Care and

Use Committee of the University of California, San Francisco. Eight-week-old C57BL/6J female mice ($n = 51$) were purchased from The Jackson Laboratory. Upon arrival, mice were randomly caught and assigned to separate cages. Mice were housed with a 12-h light/12-h dark cycle and provided food and water ad libitum. ALPHA-dri bedding and cotton string nesting material were provided in all cages. Cages of mice were then assigned to four groups: (i) control, mice ($n = 14$) were fed a standard rodent chow for the entire duration of the study; (ii) CPZ-EAE, mice ($n = 12$) received a diet containing 0.25% CPZ (bis(cyclohexanone)oxaldihydrazone; Sigma-Aldrich) mixed with standard rodent chow for a 3-week period and were returned to standard chow for the next 4 weeks. At the beginning of week 6, mice were immunized with subcutaneous injection of MOG$_{35-55}$ in complete Freund's adjuvant (Hooke Labs), followed by intraperitoneal administration of pertussis toxin on the day of immunization and the following day; (iii) CPZ-EAE + DMF (Sigma-Aldrich), mice ($n = 11$) were fed a CPZ diet and were immunized as in group (ii) and received a daily administration of DMF (100 mg/kg, in an emulsion of 0.6% Methocel) by oral gavage; and (iv) CPZ-EAE + FTY720 (Combi-Blocks, Inc), mice ($n = 14$) underwent the same experimental procedures as in group (ii) and received a daily administration of FTY720 (0.3 mg/kg, in water) by oral gavage. Both DMF and FTY720 treatments began on the day of immunization. Following immunization all animals were observed daily, and clinical signs were assessed as follows: 0, no signs; 1, decreased tail tone; 2, mild monoparesis or paraparesis; 3, severe paraparesis; 4, paraplegia; 5, quadraparesis; and 6, moribund or death. Mice reaching a score of 4 were euthanized.

**Hyperpolarization of [1-$^{13}$C]pyruvate and [$^{13}$C]urea.** [1-$^{13}$C] pyruvic acid (30.2 mg) (Sigma-Aldrich), 0.65 mg of GE-Trityl and 0.07 μL Dotarem (500 mM gadolinium-DOTA), and [$^{13}$C]urea (68.8 mg) (Sigma-Aldrich), 159 mg glycerol, 4.79 mg of trityl radical OX063 and 0.07 μL Dotarem (500 mM gadolinium-DOTA) solution were placed in the Hypersense dynamic nuclear polarization (DNP) polarizer (Oxford Instruments). The frozen sample was irradiated for 1 h and then dissolved in a solution containing 0.3 mM EDTA, 80 mM NaOH and 40 mM Tris–HCl pH 8.

**Magnetic resonance imaging.** Mice were anesthetized using isoflurane (1.5–2% in O$_2$) and a 27-gauge catheter was placed in the tail vein to allow for intravenous (i.v.) injection. Next, animals were placed in a dedicated cylindrical cradle allowing for reproducible positioning of the mouse head; which was subsequently inserted inside a dual tune $^1$H–$^{13}$C volume coil (øI = 40 mm) in a 14.1 T vertical MR system (Agilent Technologies). The respiration rate was continuously monitored through the PC-sam software interface (SA Instrument, NY, USA).

First, T$_2$-weighted images from the entire brain were acquired for adequate positioning of the grid used for hyperpolarized $^{13}$C acquisitions (repetition time 1200 ms, echo time 20 ms, slice thickness 1.8 mm, 2 averages, matrix 256 × 256, field of view 30 × 30 mm²). Hyperpolarized [1-$^{13}$C]pyruvate and [$^{13}$C]urea (0.35 mL, 80, and 78 mM, respectively) solution was injected i.v. over a period of 12 s through the tail vein catheter. From the beginning of the i.v. injection of hyperpolarized [1-$^{13}$C]pyruvate and [$^{13}$C]urea, 2D dynamic chemical shift imaging (CSI) $^{13}$C data from the brain were acquired (repetition time 60 ms, echo time 1.2 ms, spectral width 2500 Hz, 128 points, 4 s temporal resolution, flip angle 10°; matrix 8 × 8; field of view 24 × 24 mm²; slice thickness 5 mm; spatial resolution 3 × 3 × 5 mm³). Next, for T$_1$-weighted MRI, the dual tune $^1$H–$^{13}$C volume coil (øI = 40 mm) was removed and replaced by a $^1$H volume-only coil (øI = 40 mm). T$_1$-weighted images

were acquired (repetition time 120 ms, echo time 2 ms, slice thickness 0.8 mm, 10 averages, matrix 256 × 256, field of view 20 × 20 mm²) prior and after gadolinium diethylenetriamine pentaacetate (DTPA) injection (0.1 mL, 1 mmol/kg, Magnevist, Bayer).

**Magnetic resonance imaging data analysis.** Hyperpolarized $^{13}$C MRS imaging data were analyzed using the in-house SIVIC software (http://sourceforge.net/apps/trac/sivic/) and custom-built programs written in MATLAB (MATLAB R2011b, The Math-Works Inc.). The $k$-space dimensions were zero-filled by a factor of two resulting in a 16 × 16 matrix. Spectra were summed over time and a Lorentzian shape was used to fit the hyperpolarized [1-$^{13}$C]pyruvate, [1-$^{13}$C]lactate, and [$^{13}$C]urea peaks on the sum spectrum. Then, the area under the curve (AUC) of hyperpolarized [1-$^{13}$C]pyruvate, AUC of hyperpolarized [1-$^{13}$C]lactate and AUC of hyperpolarized [$^{13}$C]urea Lorentzian fits were measured for each voxel. hyperpolarized [1-$^{13}$C]lactate-to-pyruvate ratio was calculated as the ratio of the AUC. The AUC of hyperpolarized [$^{13}$C]urea signal was normalized to the AUC of hyperpolarized [$^{13}$C]urea from surrounding tissue containing blood vessels to account for variations in polarization levels and transfer time[33,44]. Next, the average from voxels containing cortex, corpus callosum, external capsule and hippocampus was calculated, and the obtained mean values were used to evaluate statistical significance between experimental groups. In addition, the mean $\mu$ and standard deviation $\sigma$ of the hyperpolarized [1-$^{13}$C]pyruvate, [1-$^{13}$C] lactate, and $^{13}$C lactate/pyruvate ratio from all animals were computed for each individual voxel. Next, the hyperpolarized [1-$^{13}$C]pyruvate, [1-$^{13}$C]lactate, and $^{13}$C lactate/pyruvate ratio values ($xi$) for each animal were normalized by converting it to a $z$-score using the following formula: $z = (xi - \mu)/\sigma$. Color heatmaps of hyperpolarized [$^{13}$C]urea, hyperpolarized $^{13}$C lactate/pyruvate ratio, and mean $z$-score for each experimental group were generated using a linear-based interpolation of the $^{13}$C 2D CSI data to the resolution of the anatomical images using custom-built programs written in MATLAB and SIVIC.

**Enzymatic assays.** Mice were transcardially perfused with ice-cold phosphate-buffered saline, and brains were rapidly dissected and snap-frozen. Samples were stored at −80 °C until further processing. Lactate dehydrogenase (LDH) and PDH activities were evaluated using spectrophotometric activity assay kits according to manufacturer's guidelines (ab102526 and ab109902; Abcam, respectively), and normalized to sample weight and protein concentration determined by the bicinchonic acid (BCA) method (ab102536; Abcam).

**Immunofluorescence analysis.** Mice were transcardially perfused with ice-cold phosphate-buffered saline followed by ice-cold 4% paraformaldehyde (PFA). Brains were dissected, fixed in 4% PFA overnight, dehydrated through a sucrose gradient (2 h at 5%, 2 h at 10%, and overnight at 20%), snap-frozen in liquid nitrogen, and kept at −80 °C until further processing. Immunofluorescence staining was performed on ten-micrometer-thick cryosections from brain using the following antibody combinations as previously described[12]: a primary chicken anti-myelin basic protein (MBP) antibody (AB9348; 1:200 dilution; Millipore) with a secondary donkey anti-chicken DyLight 549 antibody (703-506-155, 1:1000 dilution; Jackson); a primary rabbit anti-ionized calcium binding adaptor molecule 1 (Iba-1) antibody (019-19741, 1:500 dilution; Wako) with a secondary donkey anti-rabbit Alexa Fluor 555 (A31572, 1:1000 dilution; Invitrogen); a primary mouse anti-PDK1 antibody (AB110025, 1:100 dilution; Abcam) with a secondary goat anti-mouse Alexa Fluor 488 (A11017, 1:100 dilution;

Invitrogen); a primary rabbit anti-glial fibrillary acidic protein (GFAP) (AB7779, 1:500 dilution; Abcam) with a secondary goat anti-rabbit Alexa Fluor 488 (A11008, 1:1000 dilution; Invitrogen); a rabbit anti-fibrinogen (A0080, 1:200 dilution; Dako) with donkey anti-rabbit Alexa Fluor 555 (A31572, 1:600 dilution; Invitrogen); a rat anti-CD3 (MCA500G, 1: 400 dilution; BioRad) with goat anti-rat Alexa Fluor 488 (A11006, 1:800 dilution; Invitrogen); a primary rat anti-CD68 (BioRad, MCA1957, 1:100 dilution) with a secondary anti-rat AF488 (Life Technologies, A11006, 1:200).

Fluorescent widefield images were captured on an inverted Nikon TI microscope run with NIS-Elements 5.20.21 (Nikon) and equipped with an automated stage using a DS-Qi2 CMOS Camera (Nikon), a Plan Apo ×10/0.45 lens, Lambda LS lamp with filter wheel and shutter (Sutter), Lambda 10-3 controller and emission wheel (Sutter), with excitation filters 387/11x, 485/20x, 560/25x, 650/13x (Semrock) and emission filters 440/40, 525/30, 607/36, 684/24 m (Semrock) for DAPI, GFP, Cy3, and Cy5, respectively. Confocal images were captured on an inverted Nikon Ti microscope run using Micro-Manager 2.0 Gamma[45], equipped with a Zyla 4.2 CMOS camera (Andor), piezo XYZ stage (ASI), CSU-W1 Spinning Disk with Borealis upgrade (Yokogowa/Andor), Spectra-X (Lumencor), ILE 4 line Laser Launch (405/488/561/640 nm; Andor). Images were taken using a Plan Apo λ ×20/0.75 using lasers 405, 488, and 561 nm and emission filters 447/60, 525/50, 607/36, for DAPI, GFP, and RFP, respectively. Quantitative analyses of immunofluorescence images were performed using NIH ImageJ analysis software (v1.51n). CD3 cell number was evaluated using the cell counter tool. The levels of microglia/macrophages (Iba1), CD68, PDK1, fibrinogen, MBP, and GFAP were determined based on the image-covering staining and expressed in $mm^2$ or as a percentage of the total area.

**Statistics and reproducibility.** Results are expressed as mean ± s.e.m. Each dot represents an individual mouse. Statistical significance was evaluated using GraphPad Prism (v 9.1.2) with a one-way ANOVA, corrected for multiple comparisons using the Tukey HSD post-hoc test, using a Kruskal–Wallis test, corrected for multiple comparisons using the Two-stage step-up method of Benjamini, Krieger, and Yekutieli. Correlations between MR parameters and EAE scores and ex vivo tissue parameters were performed using the Pearson correlation coefficient or a simple linear regression (*$p \leq 0.05$, **$p \leq 0.01$, ***$p \leq 0.001$, ****$p \leq 0.0001$). Sample size was based on prior studies published by our group[12]. No criteria for exclusion were established a priori, and no data exclusion. No replicates were included. Potential confounders were not controlled. Experimenters were not blind to the group allocation.

**Reporting summary.** Further information on research design is available in the Nature Portfolio Reporting Summary linked to this article.

## Results

**Immunomodulatory therapies decrease disease burden in an MS mouse model.** Inflammatory demyelination throughout the mouse brain was induced by adding 0.25% cuprizone (CPZ) to the regular diet, followed by induction of experimental autoimmune encephalomyelitis (EAE) through subcutaneous immunization with $MOG_{35-55}$ peptide. Typical $MOG_{35-55}$ EAE is a disease that predominantly involves the optic nerves and spinal cord, whereas CPZ-EAE affects the whole CNS.

All in vivo and ex vivo analyses were performed at 14 ± 1 days post-immunization (dpi). Untreated CPZ-EAE mice presented tail and/or limb paralysis with a mean EAE score of 2.6 ± 0.3 (Fig. 1a). Treatment with the immunomodulatory drug DMF prevented severe disease symptoms as mice reached a mean EAE score of 1.1 ± 0.3 ($p < 0.0001$), and treatment with FTY720 completely prevented the appearance of physical impairment ($p < 0.0001$).

**Monitoring therapeutic response using in vivo hyperpolarized $^{13}C$ MRS.** After intravenous co-injection of hyperpolarized [1-$^{13}C$]pyruvate and [$^{13}C$]urea, signals from hyperpolarized [1-$^{13}C$]pyruvate, [$^{13}C$]urea and [1-$^{13}C$]lactate were observed in the brain (Fig. 1b). Hyperpolarized signals originating from areas containing the corpus callosum, external capsule, cortex, and hippocampus were averaged and used to compare experimental groups. Untreated CPZ-EAE mice displayed high hyperpolarized [1-$^{13}C$]lactate signal intensities compared to control mice (Fig. 1c). In CPZ-EAE mice treated with DMF and FTY720, lower levels of hyperpolarized [1-$^{13}C$]lactate was seen compared to untreated CPZ-EAE mice. Hyperpolarized $^{13}C$ maps showed increased hyperpolarized lactate/pyruvate throughout the brain following inflammatory demyelination, which was partially prevented by DMF and FTY720 immunomodulatory therapies (Fig. 1d). Upon quantification (Fig. 1e), we detected a 2.1 fold increase of hyperpolarized $^{13}C$ lactate/pyruvate in CPZ-EAE compared to control mice ($p < 0.0001$). Treatment with DMF and FTY720 resulted in a 1.31 and 1.35-fold decrease in hyperpolarized $^{13}C$ lactate/pyruvate, respectively ($p = 0.0331$ and $p = 0.0219$, respectively) demonstrating that hyperpolarized $^{13}C$ lactate/pyruvate can detect the effect of immunomodulatory therapies. Similar outcomes were observed when calculating the hyperpolarized $^{13}C$ lactate/pyruvate z-score (Supplementary Fig. 1a, b). The hyperpolarized $^{13}C$ lactate/pyruvate z-score in CPZ-EAE was 17.5 fold increased compared to control mice ($p < 0.0001$). Following DMF and FTY720 treatment the hyperpolarized $^{13}C$ lactate/pyruvate z-score was 1.75 and 1.97 fold decreased, respectively $p = 0.0271$ and $p = 0.0116$, respectively. We calculated the hyperpolarized [1-$^{13}C$]pyruvate and [1-$^{13}C$]lactate z-scores but did not find any significant differences between groups (Supplementary Fig. 1c, d). In contrast, no changes were detected in hyperpolarized [$^{13}C$]urea AUC between control, CPZ-EAE, and following DMF or FTY720 treatments (Fig. 1f), likely indicating no difference in brain perfusion and delivery of hyperpolarized $^{13}C$ compounds regardless of the brain pathological state.

**Monitoring therapeutic response using $T_1$ contrast-enhanced MRI.** We evaluated BBB disruption by intravenously administrating a bolus of gadolinium contrast agent solution (0.1 mL, 1 mmol/kg). We observed BBB leakiness in untreated CPZ-EAE mice and CPZ-EAE mice treated with DMF, as indicated by the appearance of hyperintense contrast in the brain (Fig. 2a). BBB leakiness was confirmed by the presence of fibrinogen in the brain parenchyma of untreated CPZ-EAE mice and DMF treated mice, which spatially corresponded to active gadolinium-enhanced MRI lesions (Fig. 2.b). Quantitative analyses further confirmed a 5.2 fold and a 3.0 fold increase in $T_1$ enhancement volume in untreated CPZ-EAE mice ($p = 0.0022$) and DMF treated mice ($p = 0.0290$) compared to control, respectively (Fig. 2.c) as well as a significant increase in fibrinogen deposition in the brain parenchyma of untreated CPZ-EAE mice and DMF treated mice (Fig. 2d, $p = 0.0029$ and $p = 0.0083$, respectively). FTY720-treated mice did not show any $T_1$ enhancement, nor fibrinogen deposition in the brain. These findings highlight that contrast-enhanced $T_1$ MRI detected the response to FT720 treatment, but not to DMF treatment, in the CPZ-EAE model.

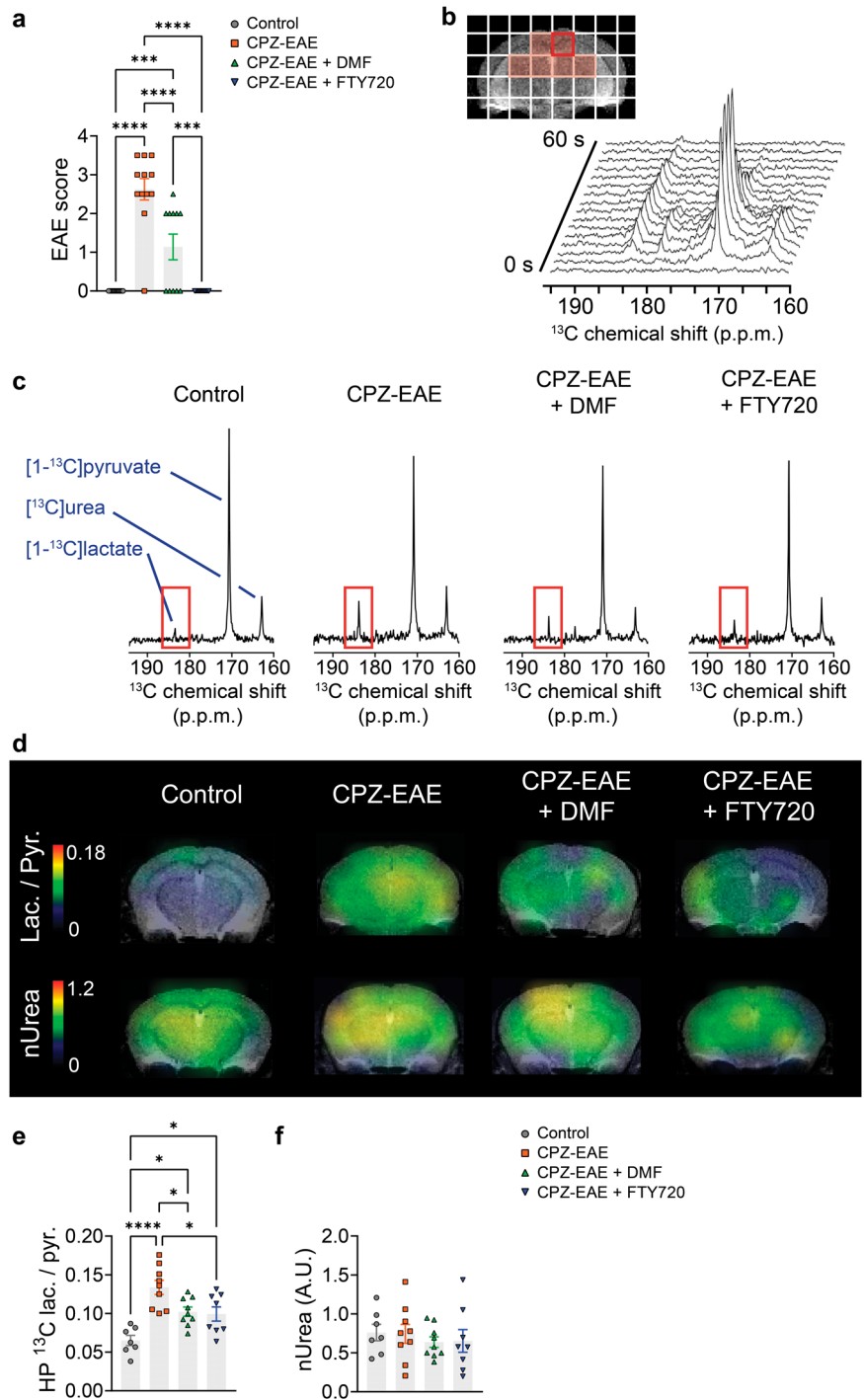

Taken together, our findings indicate that hyperpolarized [1-$^{13}$C]pyruvate conversion into [1-$^{13}$C]lactate can detect the effect of FTY720 therapy that is missed using CE $T_1$ enhanced MRI. Importantly, by detecting a lower conversion of [1-$^{13}$C]pyruvate conversion into [1-$^{13}$C]lactate despite an impaired BBB in DMF-treated mice, we further show that the hyperpolarized [1-$^{13}$C]pyruvate to [1-$^{13}$C]lactate flux does not solely appears to depend on BBB permeability.

**Enzymatic and immunohistochemical histological characterization of therapeutic response.** The activity of pyruvate dehydrogenase (PDH), the enzyme that controls pyruvate entry into the tricarboxylic cycle and its conversion into acetyl-coA, was 8.1 fold decreased in CPZ-EAE compared to untreated control brains (Fig. 3a, $p < 0.0001$), in the region comprising cortex, corpus callosum, external capsule, and hippocampus. PDH activity was 3.2 fold higher in FTY720-treated mice ($p = 0.0133$), and 3.7 fold higher in DMF-treated mice ($p = 0.0365$) compared to untreated CPZ-EAE mice. The activity of LDH, an enzyme that converts pyruvate into lactate, remained unchanged (Fig. 3b).

Immunofluorescence staining (Fig. 3c) revealed widespread inflammation in the cortex, corpus callosum, external capsule, and hippocampus in untreated CPZ-EAE mice, characterized by a 7-fold increase in Iba1$^+$ innate immune cells (Fig. 3d, $p < 0.0001$), 25 fold increase in the pro-inflammatory activation marker CD68$^+$ (Fig. 3e, $p < 0.0001$) and a 455 fold increase in CD3$^+$ lymphocytes T cells (Fig. 3f, $p < 0.0001$) compared to control

**Fig. 1 Hyperpolarized $^{13}$C lactate/pyruvate decreases following response to DMF and FTY720 immunomodulatory therapies, while [$^{13}$C]urea remains unchanged. a** EAE score at 14 ± 1 dpi. Only untreated and DMF-treated CPZ-EAE mice displayed symptoms of tail and/or limb paralysis, with decreased disease severity in DMF-treated mice. FTY720 prevented the appearance of EAE symptoms ($n = 14$ control, $n = 12$ CPZ-EAE, $n = 11$ CPZ-EAE + DMF, $n = 14$ CPZ-EAE + FTY720). **b** Representative $^{13}$C spectra from an untreated CPZ-EAE mouse brain (red voxel), after intravenous co-injection of hyperpolarized [1-$^{13}$C]pyruvate and [$^{13}$C]urea. Data were acquired every 4 s for 1 min starting from the beginning of injection. [$^{13}$C]urea peak resonance is located at 162.5 ppm; [1-$^{13}$C]pyruvate at 171 ppm and [1-$^{13}$C]lactate at 183.5 ppm. **c** Summed spectra from representative control, CPZ-EAE, CPZ-EAE + DMF, and CPZ-EAE + FTY720 mice following intravenous injection of hyperpolarized [1-$^{13}$C]pyruvate and [$^{13}$C]urea, showing increased [1-$^{13}$C] lactate peak intensity at 183.5 ppm in CPZ-EAE (red rectangle). **d** $^{13}$C lactate/pyruvate and [$^{13}$C]urea color maps obtained from $^{13}$C chemical-shift images in control, CPZ-EAE, CPZ-EAE + DMF and CPZ-EAE + FTY720 mice. Corresponding quantitative analyses from the red highlighted voxels shown in (**b**) revealed a significant increase of **e** $^{13}$C lactate/pyruvate in untreated CPZ-EAE mice. Both DMF and FTY720 treated mice displayed lower $^{13}$C lactate/ pyruvate compared to untreated CPZ-EAE mice ($n = 7$ control, $n = 9$ CPZ-EAE, $n = 9$ CPZ-EAE + DMF, $n = 8$ CPZ-EAE + FTY720). **f** [$^{13}$C]urea was unchanged between groups ($n = 7$ control, $n = 9$ CPZ-EAE, $n = 9$ CPZ-EAE + DMF, $n = 8$ CPZ-EAE + FTY720). Abbreviations: experimental autoimmune encephalomyelitis (EAE), Cuprizone and experimental autoimmune encephalomyelitis (CPZ-EAE), Dimethyl fumarate (DMF), Fingolimod (FTY720), lactate-to-pyruvate ratio (lac./pyr.), normalized urea (nUrea). Data are shown as mean ± standard error. Control is indicated by gray circles, CPZ-EAE by orange rectangles, CPZ-EAE + DMF by green triangles, and CPZ-EAE + FTY720 by blue inverted triangles (*$p \leq 0.05$, ***$p \leq 0.001$, ****$p \leq 0.0001$). Data to reproduce this figure are included in Supplementary Data 1.

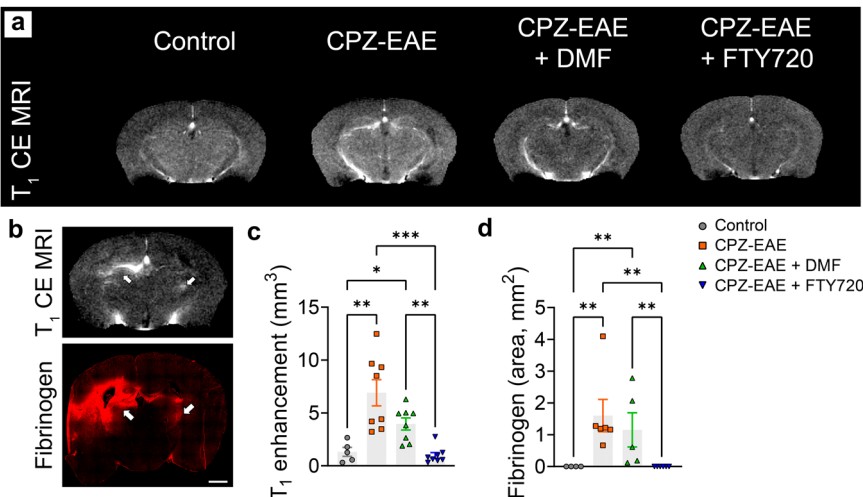

**Fig. 2 $T_1$ contrast enhancement MRI detects response to FTY720 treatment, but not to DMF therapy. a** $T_1$ weighted MR image acquired after gadolinium-DTPA injection, in which $T_1$ contrast enhancement (bright area), indicative of a leaky BBB, is clearly visible in untreated CPZ-EAE and DMF treated CPZ-EAE mouse brains, but not in FTY20 treated mice. **b** $T_1$ weighted MR image acquired after gadolinium-DTPA injection shows $T_1$ contrast enhancement (bright area, arrows), which corresponds to fibrinogen immunostaining (arrows) in the same mouse, confirming BBB leakiness indicated by MRI. Quantitative analyses revealed a significant increase of **c** $T_1$ contrast enhancement ($n = 5$ control, $n = 8$ CPZ-EAE, $n = 8$ CPZ-EAE + DMF, $n = 8$ CPZ-EAE + FTY720) and **d** fibrinogen deposition in untreated CPZ-EAE mice and DMF-treated CPZ-EAE mice, but not in FTY720 treated mice ($n = 4$ control, $n = 6$ CPZ-EAE, $n = 5$ CPZ-EAE + DMF, $n = 6$ CPZ-EAE + FTY720). Abbreviations: Contrast-enhanced magnetic resonance imaging (CE MRI), Cuprizone and experimental autoimmune encephalomyelitis (CPZ-EAE), Dimethyl fumarate (DMF), Fingolimod (FTY720). Data are shown as mean ± standard error. Control is indicated by gray circles, CPZ-EAE by orange rectangles, CPZ-EAE + DMF by green triangles, and CPZ-EAE + FTY720 by blue inverted triangles (*$p \leq 0.05$, **$p \leq 0.01$, ***$p \leq 0.001$). Scale bar is 1000 μm. Data to reproduce this figure are included in Supplementary Data 2.

mice. DMF treatment resulted in a strong decrease in innate and adaptive immune cells (Iba1[+]: −1.9 fold, $p = 0.0001$; CD68[+]: −2.5 fold, $p < 0.0001$; CD3[+]: −3.3 fold, $p < 0.0001$) compared to untreated CPZ-EAE mice. Similarly, administration of FTY720 induced an important decrease in immune cells compared to CPZ-EAE untreated mice (Iba1[+]: −1.6 fold, $p = 0.0008$; CD3[+]: −19.7 fold, $p < 0.0001$), but did not change the level of CD68[+] cells. Pyruvate dehydrogenase kinase 1 (PDK1), the enzyme that inhibits PDH activity and was previously shown to be upregulated in activated innate immune cells in an MS model, was increased in untreated CPZ-EAE mice compared to control mice (Fig. 3g, +95 fold, $p < 0.0001$). Following DMF and FTY720 treatment, PDK1 was decreased compared to untreated CPZ-EAE mice (−2.5 fold, $p < 0.0001$ and −2.9 fold, $p < 0.0001$, respectively). Reactive astrogliosis was observed in CPZ-EAE mice through an increase in GFAP staining (Fig. 3h, +3.5 fold, $p < 0.0001$). Neither DMF nor FTY720 treatments influenced

GFAP levels compared to CPZ-EAE untreated mice. Myelin basic protein (MBP) immunostaining confirmed demyelination in CPZ-EAE mice compared to control (Supplementary Fig. 2).

**Correlations analyses between MR values, disease symptoms, and ex vivo measurements.** Associations between MR values, disease symptoms and ex vivo measurements were evaluated (Table 1). Positive correlations between hyperpolarized $^{13}$C lactate/pyruvate and Iba1[+] cells ($p = 0.013$), PDK1[+] cells ($p = 0.041$), EAE scores ($p = 0.0094$), and $T_1$ enhancement volume ($p = 0.0028$) were observed. A negative correlation between hyperpolarized $^{13}$C lactate/pyruvate and PDH activity ($p = 0.014$) was found. $^{13}$C urea did not correlate with any of the parameters investigated in the study. $T_1$ enhancement volume values significantly correlated with fibrinogen immunostaining ($p = 0.027$) and EAE scores ($p < 0.0001$).

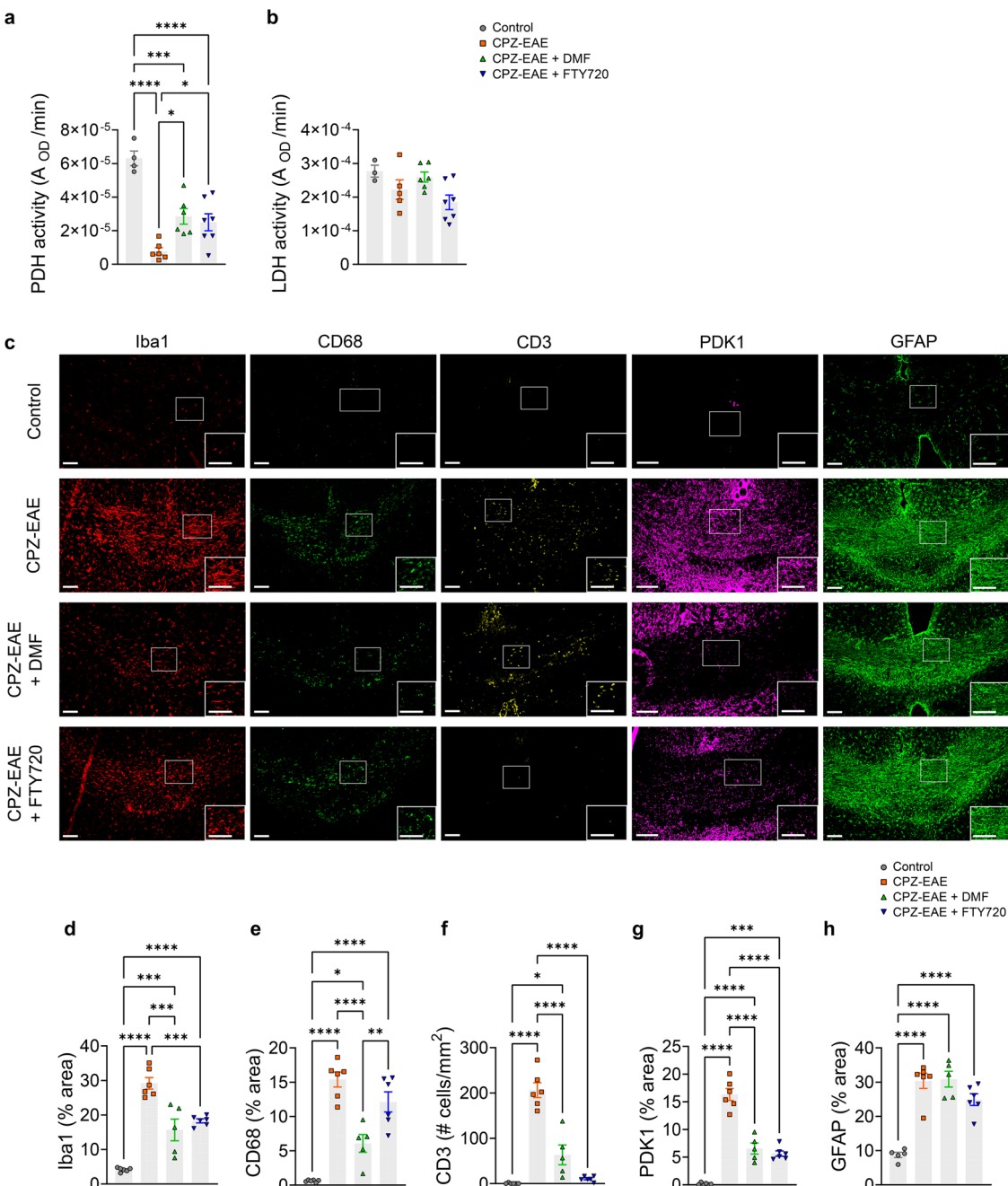

**Fig. 3 Enzymatic and immunohistochemical histological characterization of DMF and FTY720 treatment responses. a** PDH activity from control ($n = 4$ mice), CPZ-EAE ($n = 6$ mice), CPZ-EAE + DMF ($n = 6$ mice) and CPZ-EAE + FTY720 ($n = 7$ mice). **b** LDH activity from control ($n = 3$ mice), CPZ-EAE ($n = 5$ mice), CPZ-EAE + DMF ($n = 6$ mice) and CPZ-EAE + FTY720 ($n = 7$ mice). Only PDH was significantly modulated following DMF and FTY720 treatments. **c** Representative immunofluorescence staining of the corpus callosum for microglia/macrophages (Iba1, red), pro-inflammatory marker (CD68, green), T cells (CD3, yellow), PDK1 (magenta), reactive astrocytes (GFAP, green). Quantitative analyses of **d** Iba1, **e** CD68, **f** CD3, **g** PDK1, and **h** GFAP from control ($n = 5, 6, 6, 6$ and 5 mice, respectively), CPZ-EAE ($n = 5, 6, 6, 6$ and 6 mice, respectively), CPZ-EAE + DMF ($n = 4, 5, 5, 5$ and 5 mice, respectively) and CPZ-EAE + FTY720 ($n = 5, 6, 6, 6$ and 6 mice, respectively). Abbreviations: Pyruvate dehydrogenase (PDH), lactate dehydrogenase (LDH), pyruvate dehydrogenase kinase 1 (PDK1), ionized calcium-binding adaptor molecule 1 (Iba1), glial fibrillary acidic protein (GFAP), Cuprizone and experimental autoimmune encephalomyelitis (CPZ-EAE), Dimethyl fumarate (DMF), Fingolimod (FTY720). Data are shown as mean ± standard error. Control is indicated by gray circles, CPZ-EAE by orange rectangles, CPZ-EAE + DMF by green triangles, and CPZ-EAE + FTY720 by blue inverted triangles (*$p \leq 0.05$, **$p \leq 0.01$, ***$p \leq 0.001$, ****$p \leq 0.0001$). Scale bar is 100 µm. Data to reproduce this figure are included in Supplementary Data 3.

## Discussion

In this work, we report that the measurement of hyperpolarized [1-$^{13}$C]pyruvate flux towards lactate can monitor response to immunomodulatory therapies in an MS model, in line with the altered metabolic response of immune cells following treatment.

Specifically, we observed a decreased hyperpolarized $^{13}$C lactate/pyruvate following both DMF and FTY720 administration compared to untreated mice. Hyperpolarized $^{13}$C lactate/pyruvate decrease was associated with a decrease in Iba1$^+$ innate immune cells and PDK1 expression, a subsequent increase of PDH

**Table 1 Association between MR values, disease symptoms, enzyme activity, and histopathological measurements.**

| | Hyperpolarized [$^{13}$C]Lac/Pyr | Hyperpolarized [$^{13}$C]urea | T$_1$ contrast enhancement |
|---|---|---|---|
| *Disease symptoms*[a] | | | |
| EAE score | | | |
| Pearson's *r* | $r = 0.45$ | $r = -0.05$ | $r = 0.79$ |
| *p*-value | $p = 0.0094$ (**) | $p = 0.77$ | $p < 0.0001$ (****) |
| *Enzyme activity*[b] | | | |
| PDH | | | |
| Equation | $Y = -0.00074X + 0.0001$ | $Y = 0.00012X - 5e\text{-}005$ | $Y = -5.9e\text{-}006X + 5e\text{-}005$ |
| $r^2$ | $r^2 = 0.97$ | $r^2 = 0.10$ | $r^2 = 0.50$ |
| *p*-value | $p = 0.014$ (*) | $p = 0.69$ | $p = 0.29$ |
| LDH | | | |
| Equation | $Y = -0.00073X + 0.00031$ | $Y = 0.00024X + 7e\text{-}005$ | $Y = 2.03e\text{-}007X + 0.0002$ |
| $r^2$ | $r^2 = 0.30$ | $r^2 = 0.13$ | $r^2 = 0.00$ |
| *p*-value | $p = 0.46$ | $p = 0.64$ | $p = 0.98$ |
| *Immunofluorescence analyses*[b] | | | |
| Iba1 | | | |
| Equation | $Y = 330X - 16$ | $Y = -18X + 30$ | $Y = 2.9X + 7$ |
| $r^2$ | $r^2 = 0.98$ | $r^2 = 0.01$ | $r^2 = 0.60$ |
| *p*-value | $p = 0.013$ (*) | $p = 0.89$ | $p = 0.23$ |
| PDK1 | | | |
| Equation | $Y = 209X - 13$ | $Y = 9X + 1$ | $Y = 2.1X + 0.3$ |
| $r^2$ | $r^2 = 0.92$ | $r^2 = 0.01$ | $r^2 = 0.73$ |
| *p*-value | $p = 0.041$ (*) | $p = 0.92$ | $p = 0.15$ |
| CD68 | | | |
| Equation | $Y = 194X - 11$ | $Y = -16X + 20$ | $Y = 1.3X + 4$ |
| $r^2$ | $r^2 = 0.83$ | $r^2 = 0.02$ | $r^2 = 0.32$ |
| *p*-value | $p = 0.09$ | $p = 0.84$ | $p = 0.44$ |
| CD3 | | | |
| Equation | $Y = 302X - 6$ | $Y = -95X + 90$ | $Y = 2.4X + 16$ |
| $r^2$ | $r^2 = 0.81$ | $r^2 = 0.33$ | $r^2 = 0.42$ |
| *p*-value | $p = 0.10$ | $p = 0.43$ | $p = 0.35$ |
| Fibrinogen | | | |
| Equation | $Y = 21X - 1.4$ | $Y = 0.25X + 1$ | $Y = 0.3X - 0.3$ |
| $r^2$ | $r^2 = 0.59$ | $r^2 = 0.00$ | $r^2 = 0.94$ |
| *p*-value | $p = 0.19$ | $p = 0.98$ | $p = 0.027$ (*) |
| GFAP | | | |
| Equation | $Y = 296X - 5.4$ | $Y = -88X + 85$ | $Y = 2X + 18$ |
| $r^2$ | $p = 0.12$ | $r^2 = 0.28$ | $r^2 = 0.20$ |
| *p*-value | $r^2 = 0.77$ | $p = 0.47$ | $p = 0.56$ |
| Myelin | | | |
| Equation | $Y = -284X + 73$ | $Y = 66X - 1$ | $Y = -1X + 50$ |
| $r^2$ | $r^2 = 0.72$ | $r^2 = 0.16$ | $r^2 = 0.16$ |
| *p*-value | $p = 0.15$ | $p = 0.60$ | $p = 0.61$ |
| *T$_1$ contrast enhancement*[a] | | | |
| Pearson's *r* | $r = 0.54$ | $r = -0.07$ | Not applicable |
| *p*-value | $p = 0.0028$ (**) | $p = 0.73$ | Not applicable |

*PDH* pyruvate dehydrogenase, *LDH* lactate dehydrogenase, *PDK1* pyruvate dehydrogenase kinase 1, *Iba1* ionized calcium-binding adaptor molecule 1, *GFAP* glial fibrillary acidic protein, *EAE* experimental autoimmune encephalomyelitis, *Lac/Pyr* lactate-to-pyruvate ratio.
*$p \leq 0.05$, **$p \leq 0.01$, ***$p \leq 0.001$, ****$p \leq 0.0001$.
[a]Association was evaluated using the Pearson correlation coefficient.
[b]Association was evaluated using a linear regression model.

activity, thus redirecting the flux of hyperpolarized [1-$^{13}$C]pyruvate towards the tricarboxylic acid (TCA) cycle rather than towards lactate production. Importantly, these metabolic changes were observed, both in the case of a generally intact (FTY720 treated mice) or leaky (DMF treated mice) BBB, as measured by gadolinium-enhanced MRI. Altogether, our findings highlight the added value of $^{13}$C MRS imaging of hyperpolarized [1-$^{13}$C]pyruvate for the detection of active (gadolinium positive) and inactive (gadolinium negative) inflammatory lesions and demonstrate its promising potential to improve the current standard of care for diagnosis and monitoring of therapies in MS patients.

Clinical translation to MS patients is directly feasible and safe, as demonstrated by a large number of clinical studies using hyperpolarized [1-$^{13}$C]pyruvate in the human healthy brain[10,30,46] and in brain cancer patients[9,47–51] conducted worldwide in the last few years. As of December 2021, more than 780 people worldwide had received hyperpolarized [1-$^{13}$C]pyruvate injections, without any reported side effects. While the implementation of hyperpolarized $^{13}$C MRS imaging requires specialized equipment, including $^{13}$C coils and polarizers, the use of this technology is increasing rapidly. For preclinical studies, most centers are equipped with a Hypersense (Oxford Instruments) or custom-built polarizers, but novel cryogen-free dissolution dynamic nuclear polarization (dDNP) preclinical polarizers have been recently brought to the market, expanding the choice of systems available[52,53]. In the clinical setting, there are currently 25 SPINlab (GE Healthcare) hyperpolarizers being used in 13 human clinical studies across the US/Canada, Europe,

and Asia, and the number of clinical studies using this technology is steadily increasing. The achievable spatial resolution is around $1 cm^3$ currently, which compares to the average MS lesions in human patients of about $7 mm^3$. The consensus is that new sequences and ongoing hardware development will realistically lead to a $5 mm^3$ isotropic resolution in the near future. Finally, whereas only 30 voxels can be resolved in the mouse brain given its extremely small size and the need for high-field magnets, current sequences on clinical scanners can resolve up to 1,000 voxels in the human brain post-injection of hyperpolarized $[1-^{13}C]$pyruvate, thus providing information both on lesions and surrounding tissues.

FTY720 and DMF immunomodulatory therapies were selected as they differ substantially in the putative mechanism of action. FTY720, a sphingosine 1 phosphate receptor inhibitor, has been shown to primarily exerts its effect by restricting activated lymphocytes from egress out of lymphoid tissues and thus limiting their entry into the CNS[41]. DMF has anti-oxidant effects, reduces the release of inflammatory cytokines, in part mediated by the activation of the transcription factor nuclear factor (erythroid-derived 2)-like 2 (Nrf2) pathway, and decreases the number of pro-inflammatory lymphocytes subsets[42,43,54,55]. In agreement with prior studies, we found that both treatments reduced the levels of innate immune cells, that FTY720 dramatically reduced the number of infiltrating T cells, and that DMF reduced the level of proinflammatory microglia and macrophages. In addition, we observed a decreased PDK1 expression and increased PDH activity, indicating a switch in cellular metabolic activity after immunomodulatory treatment, thereby providing a tangible explanation for the observed decreased hyperpolarized $[1-^{13}C]$ pyruvate toward lactate flux. On the contrary, gadolinium-enhanced MRI was able to differentiate between FTY720 and DMF-treated groups as FTY720-treated mice did not show any gadolinium-enhanced lesions, which might be explained by the effect of FTY720 on endothelial cells[41]. Notably, in relapsing-remitting MS patients, both DMF and FTY720 have been shown to reduce the number of gadolinium-enhanced lesions[56–58]. The discrepancy between mice and patient data might be explained by the fact that DMF and FTY720 treatments were used in a prophylactic manner in our study, rather than in a therapeutic manner, which could impact disease pathogenesis differently. Importantly however, our study demonstrates that hyperpolarized $[1-^{13}C]$pyruvate was able to detect treatment effects independently of the presence of gadolinium-enhanced lesions, which is promising in terms of clinical utility for assessing therapeutic response in both lesions and normal-appearing brain matter.

Hyperpolarized $[^{13}C]$urea has demonstrated the potential to provide information about perfusion in heart, kidney, and prostate cancer, but has also been shown to not rapidly cross the BBB in healthy mice[32]. In this study, we observed no changes in hyperpolarized $[^{13}C]$urea between control, untreated, and treated CPZ-EAE groups, despite differences in BBB permeability as indicated by gadolinium-enhanced MRI and ex vivo assessment of fibrinogen deposition. This discrepancy might be explained by the fact that areas of leaky BBB are relatively small and localized, and thus remain undetected using the hyperpolarized $^{13}C$ MRS imaging sequence used for data acquisition. Another explanation might be that changes in BBB permeability may intrinsically not be detectable by hyperpolarized $[^{13}C]$urea bolus tracking. Assessment of such permeability changes might require the use of other $^{13}C$ compounds with a longer relaxation time that would enable to observe their diffusion and accumulation within the brain parenchyma. Future studies in more severe models of BBB breakdown, or following ultrasound-induced mechanical[59–61] or chemical[31,62] opening of the BBB are needed to further verify and validate the use of $[^{13}C]$urea for evaluation of BBB alterations.

Modulation of hyperpolarized $^{13}C$ lactate/pyruvate has been described in several models of inflammation, including in vitro macrophages and T cell cultures, lung injury, arthritis, myocardial infarction, graft versus host disease, traumatic brain injury, and MS[11–20]. Increased flux of hyperpolarized $[1-^{13}C]$pyruvate towards lactate has been attributed to metabolic reprogramming of immune cells upon activation[63–65], especially increased glycolysis and changes in enzymes that control the fate of pyruvate, namely LDH and PDH. One may think of LDH and PDH fluxes as having a competitive relationship, i.e. a Pasteur-like effect, which might be linked to hypoxia[66,67]. However, there is no clear demonstration of this phenomenon in MS to date. While increased LDH is consistently reported in activated macrophages and T cells[16,18,21], we did not observe any significant changes during inflammatory demyelination and following immunomodulatory treatments. In a prior study in the CPZ model, we showed that hyperpolarized $^{13}C$ lactate/pyruvate was increased in brain lesions with high levels of microglia and macrophages expressing PDK1, and corresponding to areas of decreased PDH activity. In agreement, in this study, we observed a decrease in PDH activity in untreated CPZ-EAE mice, which increased following therapies.

Correlation analyses indicated that hyperpolarized $^{13}C$ lactate/pyruvate is associated with the level of resting and activated Iba$^+$ immune cell levels, PDH activity, and PDK1 in the CPZ-EAE model. Hyperpolarized $^{13}C$ lactate/pyruvate was also associated with the EAE score and $T_1$ enhancement volume. However, we found that $T_1$ enhancement volume was associated with fibrinogen deposition only, indicating that hyperpolarized $^{13}C$ lactate/pyruvate and $T_1$ enhancement volume might reflect different underlying pathological changes. Surprisingly, no associations were found between hyperpolarized $^{13}C$ lactate/pyruvate and T cell numbers, or between hyperpolarized $^{13}C$ lactate/pyruvate and pro-inflammatory marker CD68. This result might be explained by the relatively lower number of T cells in comparison with microglia/macrophage levels and suggests that metabolic changes following innate immune cell activation may not fully mirror phenotypic changes observed with histological markers.

Monocarboxylate transporters (MCTs), which mediate the transport of pyruvate and lactate across the cell membrane, have been shown to be important modulators of the hyperpolarized $^{13}C$ lactate/pyruvate. For instance, high expression of MCT1 and MCT4 have been linked to increased hyperpolarized $^{13}C$ lactate/pyruvate in cell lines and in tumors[68–72]. It is therefore possible that changes in MCTs may also play a role in our study. Importantly, MCT expression has been shown to be differentially modulated in MS, and increased MCT1 has been noted in microglia, infiltrating macrophages, and astrocytes in active lesions[73]. In addition, MCT4 has been shown to be highly expressed in pro-inflammatory infiltrating macrophages in the EAE model[74]. Here, we did not find an association between hyperpolarized $^{13}C$ lactate/pyruvate and GFAP-expressing astrocytes, however, future studies will investigate whether MCTs expression is modulated in different cell types in the CPZ-EAE model, as well as following therapies.

Prior studies have reported that following injection of hyperpolarized $[1-^{13}C]$pyruvate, $[1-^{13}C]$lactate, $^{13}C$ bicarbonate, and in some cases $[1-^{13}C]$alanine production may be detected in the brain, although it is a matter of debate whether $[1-^{13}C]$alanine signal may arise from the surrounding tissue[29,47,75]. In our study, we were not able to detect $^{13}C$ bicarbonate or $[1-^{13}C]$alanine signals, which may be due to short $T_1$ relaxation times at ultra-high field, as well as a low signal-to-noise ratio of these metabolites in the brain. Future studies performed at lower fields would enable us to evaluate changes in $^{13}C$ bicarbonate production, which may provide a more accurate way to evaluate PDH activity.

In summary, these findings demonstrate the potential of hyperpolarized [1-$^{13}$C]pyruvate to detect metabolic changes occurring during inflammatory demyelination and to monitor the effect of immunomodulatory therapies in a preclinical model of MS. Our study provides a strong rationale to further expand the use of hyperpolarized $^{13}$C MRS technology to the clinical setting to improve the assessment of immune activation and its therapeutic modulation in MS patients, as well and as other patients with neurological disorders presenting an inflammatory component.

## Data availability

The datasets generated during and/or analyzed during the current study are available from the corresponding author on reasonable request. The source data needed to reproduce the graphs shown in Figs. 1–3 and Supplementary Figs. 1 and 2 can be found in Supplementary Data 1–5, respectively.

## Code availability

Custom computer codes or algorithms used to generate results that are reported in the paper will be made available from the corresponding author upon reasonable request.

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

## Acknowledgements

This work was supported by grants from the National Institutes of Health including a K99AI159380 (C.G.), an R01NS102156 (M.M.C.), an R21AI153749 (M.M.C., C.G.), a Cal-BRAIN Award 349087 (M.M.C.), a National Multiple Sclerosis Society research grant RG-1701-26630 (M.M.C.), a National Multiple Sclerosis Society fellowship grant FG-1507-05297 (C.G.), a Hilton Foundation—Marilyn Hilton Award for Innovation in MS Research #17319 (M.M.C.), a Dana Foundation: The David Mahoney Neuroimaging program Award (M.M.C.), an NIH Hyperpolarized MRI Technology Resource Center #P41EB013598 grant, and a UCSF Resource Allocation Program Award (M.M.C., C.G.). Imaging data for this study were acquired at the Center for Advanced Light Microscopy-Nikon Imaging Center at UCSF, including the W1-CSU Confocal obtained using an NIH S10 Shared Instrumentation grant (1S10OD017993-01A1).

## Author contributions

Conceptualization: C.G., M.M.C. Methodology: C.G., C.C., C.N. Investigation: C.G., C.C. Visualization: C.G., C.N. Funding acquisition: C.G., M.M.C. Writing—original draft: C.G., M.M.C. Writing—review & editing: C.G., C.C., C.N., A.J.G., M.M.C.

## Competing interests

The authors declare no competing interests.

## Additional information

**Peer review information** : *Communications Medicine* thanks the anonymous reviewers for their contribution to the peer review of this work. Peer reviewer reports are available.

