## [Peer Review File · Communications Medicine]

Reviewers' comments:

Reviewer #1 (Remarks to the Author):

The manuscript by Guglielmetti et al. "Imaging immunomodulatory treatment responses in multiple sclerosis using hyperpolarized ¹³C metabolic magnetic resonance imaging" describes a series of novel MRI experiments performed on a MS mouse model. The paper is timely and well written. My only reservation is based on the focus on the paper on BBB transport as a main mechanism for the observed results. Why not measure the MCT's then? I recommend a minor revision.

Minor comments:

1. The authors claims a few times, that the LDH and PDH flux is a competitive relationship i.e. a Pasteur like phenomenon. Please add extend this discussion and add some references to support this claim in MS.
2. Please add the following references and similar papers and discuss in the context of your results: DOI: 10.1002/glia.22667
3. Please add more information on the z-score used. Normally it is used for the single metabolites (lactate and pyruvate individually), how does that look? Not sure if the z-scores of the ratio is actually different from the conventional pyruvate normalization.
4. Consider using a consensus nomenclature for the correlations.
5. The T1 enhanced data has a strong correlation and I wonder what the HP vs T1 correlation was?
6. Was no ¹³C-bicarbonate measured? Please add a comment in the limitation section on this.
7. What was the spatial resolution in the CSI?

Reviewer #2 (Remarks to the Author):

Guglielmetti and collaborators have designed a very interesting approach to effectively monitoring the effects of immunomodulators in a preclinical animal model of multiple sclerosis (MS). They evaluated cell metabolism with hyperpolarized ¹³C MR spectroscopy (MRS). Many previous data suggest the feasible use of this non-invasive method in humans, too.

The work has been carefully designed and done, the results seem strong and their impact would be very relevant in the preclinical and eventually clinical environment of MS, as well as other neurological diseases in where dynamics of immunopathological basis should be easily monitored.

Methodologies are OK, including Statistics. The only recommendation of this reviewer for the publication of the present work in its current form should be a slightly more detailed explanation of the animal model chosen for their work: Why cuprizone-EAE models instead of the largely known cuprizone or EAE? They can add this in the section/s that they prefer (Introduction, Methods, Discussion) for the better comprehension of a wider audience.

Reviewer: Fernando de Castro (fdecastro@cajal.csic.es)

Reply to Reviewers' comments:

Reviewer #1 (Remarks to the Author):

The manuscript by Guglielmetti et al. "Imaging immunomodulatory treatment responses in multiple sclerosis using hyperpolarized ^{13}C metabolic magnetic resonance imaging" describes a series of novel MRI experiments performed on a MS mouse model. The paper is timely and well written. My only reservation is based on the focus on the paper on BBB transport as a main mechanism for the observed results. Why not measure the MCT's then? I recommend a minor revision.

We thank the reviewer for raising this important point and we agree that MCTs may play an important role in the observed results. Following the reviewer's suggestion, we have attempted to perform immunofluorescence staining for MCT1 and MCT4 on fixed brain sections. However, we did not get consistent and clear stains. Unfortunately, we do not have frozen samples to perform western blot analyses as all the tissue was used for LDH and PDH assays. To answer the reviewer's important comment, we thus added the following sentences to discuss our findings in relation to MCTs expression.

Page 15 Lines 313-323: *"Monocarboxylate transporters (MCTs), which mediate the transport of pyruvate and lactate across the cell membrane, have been shown to be important modulators of the hyperpolarized ^{13}C lactate/pyruvate. For instance, high expression of MCT1 and MCT4 have been linked to increased hyperpolarized ^{13}C lactate/pyruvate in cell lines and in tumors¹⁻⁵. It is therefore possible that changes in MCTs may also play a role in our study. Importantly, MCTs expression has been shown to be differentially modulated in MS, and increased MCT1 has been noted in microglia, infiltrating macrophages, and astrocytes in active lesions⁶. In addition, MCT4 has been shown to be highly expressed in pro-inflammatory infiltrating macrophages in the EAE model⁷. Here, we did not find association between hyperpolarized ^{13}C lactate/pyruvate and GFAP expressing astrocytes, however future studies will investigate whether MCTs expression is modulated in different cell types in the CPZ-EAE model, as well as following therapies."*

Minor comments:

1. The authors claims a few times, that the LDH and PDH flux is a competitive relationship i.e. a Pasteur like phenomenon. Please extend this discussion and add some references to support this claim in MS.

We thank the reviewer for this suggestion and have added the following sentences to the discussion. Page 14 Lines 293-295: *"One may think as LDH and PDH fluxes as having a competitive relationship, i.e. a Pasteur-like effect, which might be linked to hypoxia^{8,9}. However, there is no clear demonstration of this phenomenon in MS to date."*

2. Please add the following references and similar papers and discuss in the context of your results: DOI: 10.1002/glia.22667

We thank the reviewer for this suggestion and have included this key reference. We now discuss our findings in relation to this paper and additional literature (cf. reply to the first comment of the reviewer, Page 15 Lines 313-323).

3. Please add more information on the z-score used. Normally it is used for the single metabolites (lactate and pyruvate individually), how does that look? Not sure if the z-scores of the ratio is actually different from the conventional pyruvate normalization.

We calculated the z-score for the hyperpolarized $[1-^{13}\text{C}]$ lactate and $[1-^{13}\text{C}]$ pyruvate and described how we calculated the z-score in the Methods section (Page 19 Lines 400-404): “[...] the mean μ and standard deviation σ of the hyperpolarized $[1-^{13}\text{C}]$ pyruvate, $[1-^{13}\text{C}]$ lactate and ^{13}C lactate / pyruvate ratio from all animals were computed for each individual voxel. Next, the hyperpolarized $[1-^{13}\text{C}]$ pyruvate, $[1-^{13}\text{C}]$ lactate and ^{13}C lactate / pyruvate ratio values (x_i) for each animal was normalized by converting it to a z-score using the following formula: $z = (x_i - \mu) / \sigma$.”

We did not find any significant differences between the groups for the hyperpolarized $[1-^{13}\text{C}]$ pyruvate z-scores, nor for the hyperpolarized $[1-^{13}\text{C}]$ lactate z-scores. However, we noticed a trend for an increase hyperpolarized $[1-^{13}\text{C}]$ lactate z-score in CPZ-EAE mice compared to control mice ($p = 0.079$). We added the following sentences in the result section and added the hyperpolarized $[1-^{13}\text{C}]$ pyruvate and $[1-^{13}\text{C}]$ lactate z-scores in the Supplementary Figure 1 (see below).

(Page 8 Lines 155-157): “We calculated the hyperpolarized $[1-^{13}\text{C}]$ pyruvate and $[1-^{13}\text{C}]$ lactate z-scores but did not find any significant differences between groups (**Supplementary Figure 1**).”

Supplementary Figure 1. ^{13}C lactate/pyruvate mean z-score detects therapy responses.

^{13}C lactate/pyruvate mean z-score color maps overlaid on a representative T_2 -weighted image for the control, CPZ-EAE, CPZ-EAE + DMF and CPZ-EAE + FTY720 groups. Quantitative analyses revealed significant increase of the ^{13}C lactate/pyruvate z-score in CPZ-EAE mice compared to Control mice. Both DMF and FTY720 showed lower ^{13}C lactate/pyruvate z-score compared to untreated CPZ-EAE mice. No significant differences were observed between groups for the hyperpolarized $[1-^{13}\text{C}]$ pyruvate and $[1-^{13}\text{C}]$ lactate z-scores. We noted a trend for an increase hyperpolarized $[1-^{13}\text{C}]$ lactate z-score in CPZ-EAE mice compared to control mice ($p = 0.079$). (* $p \leq 0.05$, **** $p \leq 0.0001$).

4. Consider using a consensus nomenclature for the correlations.

We have made modified Table 1 accordingly (Page 34). For the analyses carried out using the Pearson correlation coefficient we report the Pearson's r , and the p-value. For the analyses carried out using the simple linear regression we report the equation, r^2 and p-value. We adjusted the text of the result section accordingly (Page 10 Lines 210-216).

5. The T1 enhanced data has a strong correlation and I wonder what the HP vs T1 correlation was?

We evaluated the correlation between T₁ enhancement volume and HP ¹³C lactate/pyruvate and found a significant correlation (Pearson r=0.54, p = 0.0028). We also evaluated the correlation between T₁ enhancement volume and HP ¹³C urea and found no correlation (Pearson r=-0.07, p = 0.73).

We added this outcome in the Results section (Page 10 Lines 210-216) and Table 1 (Page 34): *“Associations between MR values, disease symptoms and ex vivo measurements were evaluated (Table 1). Positive correlations between hyperpolarized ¹³C lactate/pyruvate and Iba1⁺ cells (p = 0.013), PDK1⁺ cells (p = 0.041), EAE scores (p = 0.0094) and T₁ enhancement volume (p = 0.0028) were observed. A negative correlation between hyperpolarized ¹³C lactate/pyruvate and PDH activity (p = 0.014) was found. ¹³C urea did not correlate with any of the parameters investigated in the study. T₁ enhancement volume values significantly correlated with fibrinogen immunostaining (p = 0.027) and EAE scores (p < 0.0001).”*

We also discuss this result in the Discussion section. Page 14 Lines 304-307: *“Hyperpolarized ¹³C lactate/pyruvate was also associated with the EAE score and T₁ enhancement volume. However, we found that T₁ enhancement volume was associated with fibrinogen deposition only, indicating that hyperpolarized ¹³C lactate/pyruvate and T₁ enhancement volume might reflect different underlying pathological changes.”*

6. Was no 13C-bicarbonate measured? Please add a comment in the limitation section on this.

We were not able to detect 13C bicarbonate. This may be due to short T₁ at high field (we use a 14.1 tesla MR scanner), as well as a low signal to noise ratio of this metabolite in the brain.

We added the following sentences in the Discussion section.

Page 15 Lines 324-331: *“Prior studies have reported that following injection of hyperpolarized [1-¹³C]pyruvate, [1-¹³C]lactate, ¹³C bicarbonate, and in some case [1-¹³C]alanine production may be detected in the brain, although it is a matter of debate whether [1-¹³C]alanine signal may arise from the surrounding tissue¹⁰⁻¹². In our study, we were not able to detect ¹³C bicarbonate or [1-¹³C]alanine signals, which may be due to short T₁ relaxation times at ultra-high field, as well as low signal to noise ratio of these metabolites in the brain. Future studies performed at lower field would enable us to evaluate changes in ¹³C bicarbonate production, which may provide a more accurate way to evaluate PDH activity.”*

7. What was the spatial resolution in the CSI?

The spatial resolution is 3 × 3 × 5 mm. We added this information and the matrix size (8 × 8) to the manuscript page 18 Line 380-381.

Reviewer #2 (Remarks to the Author):

Guglielmetti and collaborators have designed a very interesting approach to effectively monitoring the effects of immunomodulators in a preclinical animal model of multiple sclerosis (MS). They evaluated cell metabolism with hyperpolarized 13C MR spectroscopy (MRS). Many previous data suggest the feasible use of this non-invasive method in humans, too. The work has been carefully designed and done, the results seem strong and their impact would be very relevant in the preclinical and eventually clinical environment

of MS, as well as other neurological diseases in where dynamics of immunopathological basis should be easily monitored. Methodologies are OK, including Statistics. The only recommendation of this reviewer for the publication of the present work in its current form should be a slightly more detailed explanation of the animal model chosen for their work: Why cuprizone-EAE models instead of the largely known cuprizone or EAE? They can add this in the section/s that they prefer (Introduction, Methods, Discussion) for the better comprehension of a wider audience.

Reviewer: Fernando de Castro (fdecastro@cajal.csic.es)

We thank Dr. de Castro for his enthusiastic feedback and for highlighting the importance of our work in the preclinical, and eventually clinical, environment.

We have added more information regarding the choice of the animal model in the Introduction section of the manuscript (page 5 Lines 82-86).

“Although the experimental autoimmune encephalomyelitis (EAE) model is the most widely used model to study the inflammatory aspects of MS^{13,14}, the pathology mostly affects spinal cord, optic nerve and cerebellum, not the whole brain. Therefore, standard EAE is less suitable to validate novel imaging methods aiming at visualizing MS pathology in brain lesions. The cuprizone (CPZ) model is a well established model¹⁵, in which CPZ, a copper chelator, is added to the animal diet to induce oligodendrocytes death and demyelination. The location and timing of CPZ-induced lesions is highly reproducible, making this model extremely useful to validate imaging tools. However, a limitation of this model is that demyelination is not mediated by T cells¹⁶, and few T cells are present in CPZ-induced lesions. The combined CPZ and EAE (CPZ-EAE) model is a relatively recent model of inflammatory demyelination^{17,18}, characterized by reproducible induction of wide-spread demyelinated brain lesions that present parenchymal infiltration of T cells, increased number of microglia and macrophages, and reactive astrocytes, which are key features of MS pathology. Besides recapitulating key cellular aspects of MS in the brain in a reproducible manner, the CPZ-EAE model displays spinal cord pathology resulting in physical impairment linked to disease progression. Altogether, these features make it an attractive model to investigate the potential of hyperpolarized [1-¹³C]pyruvate to detect pathological changes and to evaluate the effect of immunomodulatory treatments.”

References

1. Rao, Y., *et al.* Hyperpolarized [1-(¹³C)]pyruvate-to-[1-(¹³C)]lactate conversion is rate-limited by monocarboxylate transporter-1 in the plasma membrane. *Proc Natl Acad Sci U S A* **117**, 22378-22389 (2020).
2. Granlund, K.L., *et al.* Hyperpolarized MRI of Human Prostate Cancer Reveals Increased Lactate with Tumor Grade Driven by Monocarboxylate Transporter 1. *Cell Metab* **31**, 105-114 e103 (2020).
3. Gallagher, F.A., *et al.* Imaging breast cancer using hyperpolarized carbon-13 MRI. *Proc Natl Acad Sci U S A* **117**, 2092-2098 (2020).
4. Keshari, K.R., *et al.* Hyperpolarized ¹³C-pyruvate magnetic resonance reveals rapid lactate export in metastatic renal cell carcinomas. *Cancer Res* **73**, 529-538 (2013).
5. Sriram, R., *et al.* Elevated Tumor Lactate and Efflux in High-grade Prostate Cancer demonstrated by Hyperpolarized (¹³C) Magnetic Resonance Spectroscopy of Prostate Tissue Slice Cultures. *Cancers (Basel)* **12**(2020).
6. Nijland, P.G., *et al.* Cellular distribution of glucose and monocarboxylate transporters in human brain white matter and multiple sclerosis lesions. *Glia* **62**, 1125-1141 (2014).

7. Kaushik, D.K., *et al.* Enhanced glycolytic metabolism supports transmigration of brain-infiltrating macrophages in multiple sclerosis. *J Clin Invest* **129**, 3277-3292 (2019).
8. Meiser, J., *et al.* Pro-inflammatory Macrophages Sustain Pyruvate Oxidation through Pyruvate Dehydrogenase for the Synthesis of Itaconate and to Enable Cytokine Expression. *J Biol Chem* **291**, 3932-3946 (2016).
9. Kierans, S.J. & Taylor, C.T. Regulation of glycolysis by the hypoxia-inducible factor (HIF): implications for cellular physiology. *J Physiol* **599**, 23-37 (2021).
10. Hurd, R.E., *et al.* Metabolic imaging in the anesthetized rat brain using hyperpolarized [1-¹³C] pyruvate and [1-¹³C] ethyl pyruvate. *Magn Reson Med* **63**, 1137-1143 (2010).
11. Xu, Y., *et al.* Hyperpolarized (¹³C) Magnetic Resonance Imaging Can Detect Metabolic Changes Characteristic of Penumbra in Ischemic Stroke. *Tomography* **3**, 67-73 (2017).
12. Miloushev, V.Z., *et al.* Metabolic Imaging of the Human Brain with Hyperpolarized (¹³C) Pyruvate Demonstrates (¹³C) Lactate Production in Brain Tumor Patients. *Cancer Res* **78**, 3755-3760 (2018).
13. Didonna, A. Preclinical Models of Multiple Sclerosis: Advantages and Limitations Towards Better Therapies. *Curr Med Chem* **23**, 1442-1459 (2016).
14. Cordano, C., *et al.* Validating visual evoked potentials as a preclinical, quantitative biomarker for remyelination efficacy. *Brain* **145**, 3943-3952 (2022).
15. Praet, J., Guglielmetti, C., Berneman, Z., Van der Linden, A. & Ponsaerts, P. Cellular and molecular neuropathology of the cuprizone mouse model: clinical relevance for multiple sclerosis. *Neurosci Biobehav Rev* **47**, 485-505 (2014).
16. Hiremath, M.M., Chen, V.S., Suzuki, K., Ting, J.P. & Matsushima, G.K. MHC class II exacerbates demyelination in vivo independently of T cells. *J Neuroimmunol* **203**, 23-32 (2008).
17. Ruther, B.J., *et al.* Combination of cuprizone and experimental autoimmune encephalomyelitis to study inflammatory brain lesion formation and progression. *Glia* **65**, 1900-1913 (2017).
18. Scheld, M., *et al.* Neurodegeneration Triggers Peripheral Immune Cell Recruitment into the Forebrain. *J Neurosci* **36**, 1410-1415 (2016).

REVIEWERS' COMMENTS:

Reviewer #1 (Remarks to the Author):

No further comments - I support publication in its current form.
I hope the authors will continue the investigations in this area.

Reviewer #2 (Remarks to the Author):

The authors have included now an extense paragraph with a detailed explanation on why they used EAE-cuprizone model and not EAE or cuprizone alone.
Since my point of view, the work could be published, now.